# DRL-OS: A Deep Reinforcement Learning-Based Offloading Scheduler in Mobile Edge Computing

**DOI:** 10.3390/s22239212

**Published:** 2022-11-26

**Authors:** Ducsun Lim, Wooyeob Lee, Won-Tae Kim, Inwhee Joe

**Affiliations:** 1The Department of Computer and Software, Hanyang University, 222 Wangsimni-ro, Seoul 04763, Republic of Korea; 2The Department of Computer Science and Engineering, Korea University of Technology and Education, Cheonan-si 31253, Republic of Korea

**Keywords:** computation offloading, double dueling deep Q-network, energy consumption, mobile edge computing (MEC), resource management, reinforcement learning

## Abstract

Hardware bottlenecks can throttle smart device (SD) performance when executing computation-intensive and delay-sensitive applications. Hence, task offloading can be used to transfer computation-intensive tasks to an external server or processor in Mobile Edge Computing. However, in this approach, the offloaded task can be useless when a process is significantly delayed or a deadline has expired. Due to the uncertain task processing via offloading, it is challenging for each SD to determine its offloading decision (whether to local or remote and drop). This study proposes a deep-reinforcement-learning-based offloading scheduler (DRL-OS) that considers the energy balance in selecting the method for performing a task, such as local computing, offloading, or dropping. The proposed DRL-OS is based on the double dueling deep Q-network (D3QN) and selects an appropriate action by learning the task size, deadline, queue, and residual battery charge. The average battery level, drop rate, and average latency of the DRL-OS were measured in simulations to analyze the scheduler performance. The DRL-OS exhibits a lower average battery level (up to 54%) and lower drop rate (up to 42.5%) than existing schemes. The scheduler also achieves a lower average latency of 0.01 to >0.25 s, despite subtle case-wise differences in the average latency.

## 1. Introduction

The rapid development and propagation of the Internet of Things (IoT) and smart devices (SDs) have contributed to the growth of smartphones, wearables, tablet computers, and connected devices [1,2]. SDs can be connected to wireless networks and can change and expand their usage through software applications [3]. SDs can process various mobile applications, including virtual and augmented reality, facial recognition, and online interactive games [4]. Most of these applications are sensitive to delays, perform computationally intensive tasks, and have relatively high energy consumption. Consequently, SDs with limited computation power and a small battery cannot execute these applications as intended by the developers [5,6].

In particular, wearable device hardware must be compact for portability. This requirement limits the device features and performance required to run certain applications. Therefore, mobile devices process data generated by auxiliary devices within a margin of error, such as cameras and heart rate monitors. For example, even high-performance smartphone processors cannot simultaneously handle multiple resource-intensive applications for extended periods without throttling the hardware.

To resolve this problem, resource-intensive computational tasks can be transferred to a processor or an external server with larger computational resources, an approach known as computation offloading. Initially, these tasks could be addressed with mobile cloud computing (MCC) because most of them were compute-intensive tasks. However, the number of delay-sensitive tasks requiring real-time processing, which were difficult to solve with MCC, have increased over time. In [7], the use of mobile edge computing (MEC), also known as multi-access computing [8] was proposed for efficient task processing. Unlike MCC, MEC provides an effective solution for computation offloading by placing a small, high-performance server near smart device users (SDUs) in a distributed network. SDUs can offload computing tasks to the MEC server connected to the base station through a wireless network. Computation offloading improves the SDU’s quality of experience (QoE) of applications by significantly reducing the latency and energy consumption. These merits have increased research interest in the computation offloading of MEC systems.

For example, unmanned aerial vehicle (UAV) [9] services in a smart city, traffic data analysis, delivery, and public safety networks can significantly reduce delays with MEC as they gather large volumes of data that require real-time processing. Therefore, MEC is a promising paradigm that can support delay-sensitive services and smart applications. However, despite its potential, MEC has some drawbacks. First, MEC is not as effective for compute-intensive applications because it has a lower computing performance than MCC [10]. The processing capacity allocated to an SD by MEC varies with workload because the MEC may have limited processing capacity. A high load may be generated if many SDs offload tasks to a single MEC server. SDs offload tasks to reduce processing delay and save energy consumption; however, a high load may cause a task offloading decision conundrum.

Studies have proposed task offloading algorithms to solve this problem, for example Wang et al. [11] suggested an algorithm that made decisions about offloading of mobile devices to maximize the benefits in terms of network. Ali et al. [12] proposed an offloading algorithm that selected a set of optimal components based on the energy consumption, network condition, computation load, and data volume according to the residual energy of the mobile device and the application components. Both [11,12] suggested methods to process tasks by offloading them to an MEC server for cases where mobile devices had limited resources. If the mobile device has several tasks to process or if multiple users request processing simultaneously in an MEC environment, the device may be subjected to a high load, resulting in a lengthy delay in processing tasks. Furthermore, they did not consider scenarios in which one or more tasks may be terminated as a result of a deadline expiring. Tang et al. [13] proposed an algorithm that allowed each device to determine which task to offload, given that the processing of tasks with expired deadlines may be terminated if many devices offloaded tasks to the MEC, considering the load of a single MEC server. However, they did not consider the energy cost of an SD, although the latency was considered when tasks were input to the queue system of an SD and dropped during processing. The approach described in [14] aimed to minimize a task latency and conducted rule-based decision-making on whether to offload to the MEC server by considering the task buffer queue, the operation status of the local device, and the transmission queue. We focused on the computation offloading scheduling problem from the perspective of SDs. The contributions of this study for more realistic MEC scenario processing of computation-intensive and delay-sensitive SD tasks are as follows:(1)Task segmentation is assumed to be a complex process because simple task segmentation may not be realistic owing to the dependency between bits in the task. We consider the default queue system in the SD and MEC. Each task can be processed by core, and processed as many as the number of multi core.(2)Unlike previous studies that focused on delay tolerance, this study considers delay-sensitive tasks with a variable deadline. This is because a task will consume energy or will be delayed once task processing is initiated, even if it does not meet the deadline or cannot be completed due to lack of energy.(3)Successful processing, latency, and energy of the task should be considered for local and remote processing. If it is difficult to process the task, the SD drops the task without processing it.(4)The system is highly complex because it has several variables that need to be considered to reflect realistic scenarios. Therefore, we propose the deep-reinforcement-learning-based offloading scheduler (DRL-OS) approach. The DRL is used to obtain the task offloading policy from the information of the task to be processed in an SD, network, and MEC state.

The rest of this paper is organized as follows. Section 2 introduces related works and investigates previous research and reinforcement learning (RL). Section 3 describes the mathematical models for components, local computing, and offloading. Section 4 proposes a DRL-based offloading scheduler and explains its structure and operation. Section 5 presents the performance evaluation result. Finally, Section 6 summarizes the study and suggests future research directions.

## 2. Related Work

In this section, studies related to offloading and reinforcement learning are reviewed.

### 2.1. Computation Offloading

Many studies on computation offloading have been conducted over the last few years to improve SD performances. Each proposed offloading technology has a different purpose and method; therefore, we only explore research related to our study. Studies on offloading can be classified into latency-, energy-, and cost-based offloading.

The latency-based offloading method aims to minimize the time consumed by the processing of delay-sensitive applications. Jia et al. [15] developed a heuristic program segmentation algorithm in an MCC framework to use the method based on the concept of load balancing between a mobile device and a server to minimize system delay. However, they did not consider energy consumption. Sun et al. [16] suggested a latency-aware workload offloading strategy in terms of a new cloudlet network to minimize waiting time. Through this, mobile users minimized the average response time by offloading the computation-intensive tasks of applications to an appropriate cloudlet. Samanta et al. [17] considered both delay-tolerant and delay-sensitive tasks to achieve optimized service delay and revenue. Furthermore, to provide a better QoE for delay-sensitive applications, Tang et al. [13] formulated the offloading problem to minimize the expected long-term cost (latency and deadline), while considering delay-sensitive tasks in a total offloading situation. In summary, these studies focused on latency without considering energy consumption.

The energy-based offloading method aims to reduce the energy footprint by identifying the cause of energy-sensitive tasks, considering the battery of a device with portability constraints. Xiang et al. [18] proposed an energy-optimal mobile computing framework to process applications by locally optimized energy or through offloading. However, if the task of the application exceeded the deadline constraint, it was processed locally. Zhang et al. [19] developed an expandable dynamic programming algorithm that selects the adaptive LTE/Wi-Fi link and integrates data transmission schedules to reduce the total energy consumption of mobile devices in an MCC system. Zhang et al. [20] developed an optimal computation offloading algorithm for mobile users in intermittently connected cloudlet systems and formulated the Markov decision process (MDP) model to minimize the energy cost.

Guo et al. [21] used the variability of features of mobile devices and user preferences to research an efficient energy computational offloading management method. The energy consumption of each User Equipment (UE) was minimized by commonly optimizing the offloading decision, spectrum, power, and resource allocation in an MEC system. Yang et al. [22] formulated the offloading energy optimization problem, which considered the computation capabilities and service delay requirements needed to improve the efficiency of the total energy consumption in every system entity. The study did not consider time costs, but considered the energy consumption of MEC and MCC through a power grid.

Meanwhile, other studies have attempted to reduce the energy cost (i.e., improve efficiency) by considering both waiting time and energy. In [23], a joint communication and computation resource allocation algorithm was proposed to minimize the energy consumption of mobile devices, while guaranteeing the offloading waiting time requirements in a multi-cell MEC system. They investigated the joint task offloading and resource allocation problem by considering both the required execution time and energy consumption. Lyu et al. [24] programmed an algorithm that minimized the offloading energy consumption according to the task deadline, focusing on delay-sensitive tasks. Eshraghi et al. [25] proposed an algorithm that optimized the decision on the allocation of computational resources of the offloading and MEC of mobile devices, considering their uncertain computation requirements. Yang et al. [26] developed a distributed offloading algorithm to resolve competition for wireless channels among mobile devices. The objectives of computation offloading in MEC are minimizing energy consumption and processing tasks within the deadline constraints. The study focused on the processing aspect between MEC and MCC.

This study considered offloading system scenarios that reflected realistic situations. In [14], tasks were considered to be divisible, but this may be unrealistic owing to the interdependency of bits in tasks. In [15,16], situations where tasks could not be divided but did not consider queue systems were shown. Furthermore, [24] considered fixed task deadlines. However, our study applied realistic variable deadlines. To summarize the contents of the literature above, computational offloading papers are listed in Table A1 of Appendix A.

Unlike previous studies, this study considered more realistic and practical scenarios using existing research on the deadline and task drop point. Furthermore, we proposed an effective computation offloading scheduler that considers delay, energy consumption, and drop rate through the task scheduling and offloading decision based on DRL.

### 2.2. Types of Reinforcement Learning

#### 2.2.1. Reinforcement Learning

RL is a machine learning method for agents that determines which action can receive the maximum reward in a given situation. Agents attempt to learn the policy π that can obtain the maximum reward in the future through interaction with the environment [27]. Policy π is a table that maps the action *a* to be performed by the agent in state *s*. In this scenario, the agent executes an action through continuous interaction with the environment and can achieve a reward that corresponds to that action. The objective of this type of learning is to maximize the sum of rewards. As shown in Figure 1, when the agent selects action $a_{t}$ according to policy π in state $s_{t}$, the environment yields the next state $s_{t+1}$ and reward $r_{t}$ to the agent. Then, policy π is updated based on the reward provided in the next environment. Most RL problems are defined as an MDP [28]. The MDP is a decision-making model based on the Markov chain. This model satisfies the Markov property, which is unrelated to the history up to the present state but is only influenced by the probability of the previous state. To solve the main problem of this study, it must be converted to an MDP type. In general, the MDP model is defined by the tuple (S, A, P, R, γ), where *S* and *A* are the state and action spaces, respectively. *P* is the state transition probability distribution that indicates the probability of the current state moving to a different state according to a specific action. *R* denotes the reward that is immediately received from the environment after the action is completed, and γ∈[0, 1] is a discount factor that is the sum of the discount rewards of the goal and is defined as follows:(1)$$G_{t}=\overset{n}{\underset{i=0}{\sum }}\gamma^{i}r_{t+i}$$
where $r_{t+i}$ is the reward in each time step *t + i*.

#### 2.2.2. Q-Learning

Q-learning is a model-free learning method [29], which approximates the *Q* value for a state–action pair and uses this value to determine the action to be executed in a specific state. This algorithm is used to compose the *Q* table according to the state–action pair. Here, the *Q* table is a simple lookup table. When an action is taken in a specific state, the value of *Q* represents the value for this action and is stored in the *Q* table. The action that has the largest *Q* value must be selected. Q-learning is a process of creating the *Q* table for policy decision making, and the equation for evaluating the value of the action is defined as
(2)$$Q\left(s_{t},a_{t}\right)=r_{t}+\gamma \underset{a_{t+1}}{\max}Q\left(s_{t+1},a_{t+1}\right)$$
where $s_{t}$ and $a_{t}$ are the state and action at a particular time step, respectively. For a specific step *t*, the action $a_{t}$ is selected first according to the current state $s_{t}$ and *Q* table. Subsequently, reward $r_{t}$ and the next state $s_{t+1}$ are obtained from the environment. Consequently, the new $Q\left(s_{t},a_{t}\right)$ is updated as follows:(3)$$Q\left(s_{t},a_{t}\right)\leftarrow Q\left(s_{t},a_{t}\right)+\alpha \left(r_{t+1}+\gamma \underset{a_{t+1}}{maxQ}\left(s_{t+1},a_{t+1}\right)-Q\left(s_{t},a_{t}\right)\right)$$
where $Q\left(s_{t},a_{t}\right)$ on the left side represents the updated value of $s_{t}$ and $a_{t}$, whereas $Q\left(s_{t},a_{t}\right)$, $r_{t+1}$ and $\underset{a_{t+1}}{maxQ}\left(s_{t+1},a_{t+1}\right)$ represent value and the maximum expected future reward value, respectively. Further, γ and α∈(0, 1) represent the discount factor and learning rate, respectively. This procedure is repeated until the terminal state is reached.

#### 2.2.3. Deep Q-Network

Conventional Q-learning saves the action value in the *Q* table. However, it is impossible to create all *Q* tables in the MEC environment and conditions considered in this study due to many combinations of state and action. Therefore, we used the deep Q-network (DQN), which uses a neural network to approximate the *Q* value. A previous study demonstrated the improved performance and learning advantages of the DQN [30]. The DQN algorithm efficiently predicts the *Q* value using a neural network that has the parameter θ instead of *Q* table [31]. The agent that uses a neural network that has learned the *Q* value is called the Q-network. The approximation of the *Q* value for an action *a* selected in a specific state $s_{t}$ using the Q-network is expressed as *Q*($s_{t}$, $a_{t}$, *θ*). Parameter θ refers to the weight of a neural network, and the Q-network is trained by updating θ in each iteration so that it approaches the real *Q* value. The DQN is trained toward the target value by minimizing the loss in each iteration. The loss function equation is defined as
(4)$$L\left(\theta \right)=E[{\left(r_{t+1}+\gamma \underset{a_{t+1}}{\max}Q\left(s_{t+1},\;a_{t+1};\theta^{-}\right)-Q\left(s_{t},\;a_{t};\theta \right)\right)}^{2}]$$
where $Q\;\left(s_{t+1},\;a_{t+1};\theta^{-}\right)$ is the largest among the target *Q* values predicted through the target network with $\theta^{-}$ as the parameter, and $Q\left(s_{t},\;a_{t};\theta \right)$ is the *Q* value predicted in the evaluation network with θ as the parameter.

#### 2.2.4. Double Deep Q-Network

After selecting an action, the DQN uses the largest of the available action *Q* values in the next state when evaluating that action. However, this can lead to overestimation of the *Q* value. To solve this problem, the double DQN [32] separates the target Q action selection and calculation. It finds the action $a_{t+1}$ of $Q\;\left(s_{t+1},\;a_{t+1};\theta^{-}\right)$ in the target network to select the maximum *Q* value. Next, the *Q* value of this action is calculated in the evaluation network. Double DQN avoids overestimation from the selection of the maximum action because the target network divides the networks for selecting an action and for evaluation. The target *Q* value of the double DQN can be updated as follows:(5)$$Q\left(s_{t},a_{t}\right)=r+\gamma Q(s_{t+1},\;arg\underset{a_{t+1}}{\max}Q\left(s_{t+1},\;a_{t+1};\theta^{-}\right);\theta )$$

#### 2.2.5. Dueling Deep Q-Network

The DQN is slow because it requires the combination of several state actions for training. To solve this problem, the dueling DQN [33] does not need to search the combinations of all states and actions. Instead, it estimates all actions in one search. Hence, while the conventional method requires a long time as it has to search all actions, the dueling DQN has a shortened training time because it requires fewer searches. Figure 2 (top) illustrates the conventional DQN model. In Figure 2 (bottom), the dueling DQN structurally resembles the first part of the DQN. However, in the second half, the dueling DQN maps the output to two fully connected layers and merges them into one *Q* value. It obtains the state action value by merging the two components, i.e., value function *V*($s_{t}$) and advantage function *A* ($s_{t},{\;a}_{t}$), as follows:(6)$$Q\left(s_{t},\;a_{t}\right)=V\left(s_{t}\right)+A\left(s_{t},\;a_{t}\right)$$
where $V\left(s_{t}\right)$ is the expected value of the reward that can be obtained in a specific state s, and $A\left(a_{t}\right)$ is a value indicating the relative importance of a specific action. The conventional DQN calculates every *Q* value for each action, even for a state in which it is undesirable for the agent to be rewarded. However, if training is conducted with the value and advantage functions, the valuable state can be known even if the agent does not select every action in each state for training; thus, the training time can be reduced. The equation for calculating the *Q* value of the dueling DQN is as follows:(7)$$Q\left(s_{t},\;a_{t};\theta \;\;\right)=V\left(s_{t};\;\theta \right)+A\left(s_{t},\;a_{t};\theta \right)-\frac{1}{\left|A\right|}\underset{a_{t+1}}{\sum }A\left(s_{t},\;a_{t+1};\theta \right)$$

## 3. System Model

In this section, we present the MEC-based network architecture and propose a system model for local and offloading processing of tasks in an SD based on the architecture. First, details of the analysis (measurement of degree, metric), including the components, calculation model, and energy consumption adopted in this study, are explained. In addition, the minimization problem for the weighted sum of latency and energy is formulated. Table 1 lists the main notations.

### 3.1. Components

As shown in Figure 3, the MEC server is a computing device installed at a wireless base station. We assumed a two-tier computational offloading model in which SDs are connected to one MEC server through 5G or Wi-Fi [34,35]. However, the SDs were assumed to be fixed or have very low mobility. Here, one or more wearable devices were connected to the SDs through short-distance communication technology. Furthermore, the wearable devices relay the generated computation-intensive tasks to the SDs for processing. The set of SDs *M = {1, 2,…, m}* processes tasks received from the wearable devices locally, offloads them to a remote MEC server through a base station (BS) or drops if it is impossible to process them. Here, we assumed that each mobile device has the same processing performance. In this study, offloading is not expanded to MCC or another MEC, and only the MEC of a single-layer structure is considered. When the task received from an SD is processed, the MEC transmits the result to the corresponding device. However, the size of the computational result is negligible, which corresponds to several computing scenarios such as facial recognition and video analysis [4,36]. Therefore, this study does not consider the cost of the result response when offloading is decided.

The set of tasks $\tau_{i}=\left\{1,\;2,\ldots ,\;i\right\}$ is generated in wearable devices by default and unconditionally transferred to mobile devices. We assumed that tasks cannot be processed separately, and that they can be transmitted and processed only in the unit of size in which they were first generated. To explain the tasks that have arrived at a specific SD, each task was defined as $\Lambda_{i}=\left\{K_{i},\;D_{i}\right\}$, where $K_{i}$ and $D_{i}$ denote the data size and deadline, respectively. The processing time of task $\Lambda_{i}$ must not exceed the deadline, regardless of whether the task is processed locally or by offloading. If the processing time exceeds the deadline, the task is dropped. When a new task $\Lambda_{i}$ arrives at the SD, where m∈M, the DRL-based scheduler allocates this task to either the processing or transmission queue. Figure 4 depicts the DRL scheduler and the queue. The assumptions and their impact are summarized as follows:

It is assumed that the SDs are connected to one MEC server through 5G or Wi-Fi. SDs offload tasks to the wireless network.We assumed that the SD is either fixed or has low mobility. Some of them are sensor network devices or are used by users resulting in low mobility.Each SD was assumed to have the same processing performance. The number of WDs connected to the SD and the task size are different. Therefore, even if the performance of the device is the same, different results can be obtained due to external influences.It was assumed that tasks could be processed and transmitted only in the first created unit of size. In addition, task segmentation is assumed difficult because of the dependency between bits in the task. Therefore, the task is sent with full offloading.

### 3.2. Computation Model

Task $\Lambda_{i}$ can be processed in the MEC server by the local computation resource of an SD or by offloading. These models are termed as “local computing” and “remote edge computing,”, respectively, and are detailed below.

#### 3.2.1. Local Computing

The prediction value $\hat{y}=0$ and task $\Lambda_{i}$ of the DRL scheduler were processed locally. Here, the tasks that were already stored in the processing queue were processed first, followed by the latest tasks. Each SD has a computational capability (CPU cycles per second), which is denoted as $F^{loc}$. The $cycle_{i}$ denotes the required number of cycles per bit of $task_{i}$, and this value may vary with the task type. Therefore, the local computing processing time, $L_{i}^{loc}$, required to run $K_{i}$ bytes is expressed as follows:(8)$$L_{i}^{loc}=\frac{\left(\delta^{proc}+K_{i}\right)cycle_{i}}{{ℱ}^{loc}}$$
where $\delta^{proc}$ is the processing queue of the device.

According to Guo et al. [37], the CPU power consumption of a specific SD, $P^{loc}$, is a super linear function of ${ℱ}^{loc}$. It is a unique feature that changes with the SD and is defined as
(9)$$P^{loc}=\varsigma^{loc}core^{loc}{\left({ℱ}^{loc}\right)}^{2}$$
where $\varsigma^{loc}$ is the effective switching capacity set to $10^{-9}$ [38], and $core^{loc}$ is the number of CPU cores. The energy consumption of the SD, $E^{loc}$, can be expressed in terms of the CPU power consumption and task processing time as follows:(10)$$\begin{array}{ll} E^{loc} & =\;P^{\mathrm{loc}}L_{i}^{\mathrm{loc}}\\ & =\varsigma^{\mathrm{loc}}{\mathrm{core}}^{\mathrm{loc}}{ℱ}^{\mathrm{loc}}\left(\delta^{\mathrm{proc}}+K_{i}\right)cycle_{i} \end{array}$$

#### 3.2.2. Remote Edge Computing

When the prediction value of the DRL scheduler was $\hat{y}=1$, the SD *m* processed task $\Lambda_{i}$ through offloading. After processing a task, the MEC server returned the computation result to the SD *m*. However, because the result data size was small and the downlink transmission rate was high, the transmission time and energy required to relay the computation result from the MEC server to the SD was dismissed. Thus, the total processing time of task $\Lambda_{i}$ comprises two parts: the time taken to transmit task $\Lambda_{I}$ from the SD *m* to the MEC; and the processing time of the MEC server. First, the transmission time for task $\Lambda_{i}$ from the SD to the edge node, $L_{i}^{tr}$, is defined as
(11)$$L_{i}^{tr}=\frac{\delta^{tran}+K_{i}}{DataRate^{CT}}$$

The SD first transmits the tasks waiting in the transmission queue, followed by the newly input tasks. $\delta^{tran}$ is the size of the tasks that first arrive and await transmission, and $DataRate^{CT}$ is the transmission rate. For $DataRate^{CT}$, the transmission rate of 5G or Wi-Fi was used, depending on the communication type to which the SD is currently connected. The processing time $L_{i}^{MEC}$ for processing task $\Lambda_{i}$ that has been offloaded from an SD to the MEC server can be expressed as follows:(12)$$L_{i}^{MEC}=\frac{\left(\delta^{tran}+K_{i}\right)Cycle_{i}}{{ℱ}^{MEC}}$$
where ${ℱ}^{MEC}$ is the computational capability of MEC. Therefore, the total processing time for offloading $L_{i}^{ofl}$ is defined as
(13)$$L_{i}^{ofl}=L_{i}^{tr}+L_{i}^{MEC}$$

Furthermore, the energy consumption of the SD when task $\Lambda_{i}$ is offloaded to the MEC server is expressed as
(14)$$E^{ofl}=P^{tr}L_{i}^{tr}$$
where $P^{tr}$ is the transmission power of the SD. In this study, the computing energy of the MEC server was not considered because it was provided by a wired power grid. Furthermore, we assumed that the MEC distance to every SD was the same. Therefore, the transmission power was also identical.

## 4. DRL-Based Offloading Scheduler

In this section, we introduce the DRL-OS. We formulated the task offloading decision-making of an SD as an MDP. To minimize the latency, energy consumption, and drop rate, each SD gauged the state using RL and selected a mode for task processing. We defined the state space, action space, and reward for offloading decisions.

### 4.1. State Space

Each state in the state space is composed of various pieces of information. In this model, each state is composed of seven pieces of information: task size (unit: bytes), deadline (unit: ms), processing queue of the SD, transmission queue and MEC processing queue information (byte size of information), battery level, and communication type (5G or Wi-Fi). Thus, the state is defined as follows:(15)$$S=\left\{K,\;D,\;\delta^{proc},\delta^{tran},\delta^{MEC},B^{r},CT\right\}$$
where *K* is the size of the task, *D* is the task deadline, and the $\delta^{proc},\;\delta^{tran}\;and\;\delta^{MEC}$ represent the processing queue of devices, the transmission queue of devices and the processing queue of MEC, respectively. Further $B^{r}$ and CT represent the residual batter of SD and transmission rate of the SD, respectively.

### 4.2. Action Space

The action spaces of this study comprise three actions, and can be defined as
(16)$$A=\left\{a_{0},\;a_{1},\;a_{2}\right\}$$
where $a_{0}\left(t\right)$ denotes local processing, $a_{1}\left(t\right)$ denotes remote processing in the MEC, and $a_{2}\left(t\right)$ denotes the action that drops the task. In the case where both local and MEC can be processed, decision on the action should align with the lower cost of either processing time or energy consumption. If neither selection is possible, the task should be dropped. When the processing time breaches the deadline or the energy of the SD is insufficient, $a_{2}$ should be selected.

### 4.3. Reward

In this section, the cost and reward are numerically defined according to local and remote processing and dropping the task. To run task $\Lambda_{i}$ according to each action, a decision must be made between local and offloading. The cost of local computing $C_{t\;}^{loc}$ is defined as
(17)$$C_{t\;}^{loc}=\omega L_{t}^{loc}+\left(1-\omega \right)E_{t}^{loc}+\Delta T_{t}^{loc}$$
where ω is a weighting coefficient to balance processing time and energy consumption and $\Delta T_{t}^{loc}$ is the time excess cost penalty.

For SDs with sufficient battery capacity to process data through both local and remote connections, Equation (17) is used to select a small action that requires less cost value. If the device’s battery does not have sufficient charge, the task is dropped. The cost of the dropping action is defined as $C_{t}^{dr},$ and its value is 1. When the processing time breaches the deadline, regardless of whether the battery has sufficient charge, the algorithm imposes $\Delta T_{t}^{loc}$. If this value is very large, the task is dropped. In general, a minimal timeout value does not affect the Quality of Service (QoS) because it is designed to apply time tolerance later. However, as time tolerance varies with the type of data, the case in which the deadline is breached is designed to be dropped from the simulation process. The penalty $\Delta T_{t}^{loc}$ for breaching the deadline is defined as
(18)$$\Delta T_{t}^{loc}=\{\begin{array}{cc} 0 & if\;L_{t}^{loc}\leq D_{t}\\ {\left(L_{t}^{loc}-D_{t}\right)}^{2} & if\;L_{t}^{loc}>D_{t} \end{array}$$

When processed with the deadline, $\Delta T_{t}^{loc}$ becomes 0 and does not affect the cost. However, if the deadline is exceeded, it is designed to be added to the form of a square of two that is considerably influenced by the larger timeout value. If the absolute value is less than 1 due to the characteristics of the square, the value becomes negligible. Therefore, the difference between less than 1 s may not be effective in imposing the penalty. To prevent this, we process the time calculation in milliseconds only in this case. For example, if the time difference is 0.1 s, the value of $\Delta T_{t}^{loc}$ is 10,000 ms, not 0.01. Similarly, costs $\;C_{t\;}^{ofl}$ and $\Delta T_{t}^{ofl}$, which handle task $\Lambda_{i}$ as an offloading task are defined using Equations (19) and (20), respectively.
(19)$$C_{t}^{ofl}=\omega L_{t}^{ofl}+\left(1-\omega \right)E_{t}^{ofl}+\Delta T_{t}^{ofl}$$
(20)$$\Delta T_{t}^{ofl}=\{\begin{array}{cc} 0 & if\;L_{t}^{ofl}\leq D_{t}\\ {\left(L_{t}^{ofl}-D_{t}\right)}^{2} & if\;L_{t}^{ofl}>D_{t} \end{array}$$
where $L_{t}^{ofl}$ and $E_{t}^{ofl}$ are the total processing time and energy consumption for offloading task $\Lambda_{i}$, respectively.

Each agent receives an immediate reward $r_{t}$ for action $a_{t}$ selected from time slot *t*. In general, the reward function is related to the cost function. The goal of optimizing the proposed problem is to minimize costs. The local computing cost $C_{t\;}^{loc}$, offloading computing cost $C_{t}^{ofl}$, and cost of dropping a task $C_{t}^{dr}$ are defined as
(21)$$reward_{t}=\{\begin{array}{cc} -C_{t\;}^{\mathrm{loc}} & if\;a_{t}=0\\ -C_{t\;}^{\mathrm{ofl}} & if\;a_{t}=1\\ -C_{t\;}^{\mathrm{dr}} & if\;a_{t}=2 \end{array}$$

Therefore, this study aims to minimize the costs while maximizing the rewards.

### 4.4. Architecture of DRL-Based Offloading Scheduler

The action in the current state in a simulation environment was estimated using the DRL-based computation offloading scheduler with double dueling DQN (D3QN) [39]. D3QN has advantages in convergence and stability [40]. The RL used in this experiment is the D3QN algorithm. To implement the neural network, a fully connected deep neural network (DNN) comprising one input layer, two hidden layers, and one output layer was used. The first three layers have 256, 256, and 128 neurons, respectively. The fourth neural layer was divided into advantage (action advantage) and value (state value) functions, which originated from the dueling DQN. In the last layer, the advantage and value functions are merged into the *Q* value. The model was trained to learn the optimal policy by the offline method. After training the model, we performed decision making to process tasks in the DRL-based scheduler of the SD.

This model has a greater degree of complexity than conventional scenarios that only consider computing and communication, because it considers various parameters such as computing, energy, and deadline to optimize the latency, energy, and task drop rate. The model encounters numerous system states as it considers a realistic scenario with a dynamic task size, deadline, processing queue, SD transmission queue, MEC server processing queue, SD battery level, and communication type. Furthermore, decisions should be made regarding the method to process tasks by reflecting the current state in the SD.

In general, conventional optimization methods such as convex optimization and game theory cannot seamlessly execute the optimal decision in a probabilistic environment. However, RL searches for the optimal strategy without prior knowledge through interaction with the environment. As mentioned, this study proposes the DRL-based computation offloading scheduler based on the D3QN, which combines double and dueling DQNs.

#### 4.4.1. D3QN Architecture

Figure 5 outlines the network structure of the D3QN scheduler. First, the input layer contains the state information of each SD. Each state is input in the form of an input vector to the D3QN network and delivered to the evaluation and target networks. Overestimation by the conventional DQN algorithm causes overestimation of the *Q* value and performance deterioration when selecting the action. Hence, action selection can be markedly improved by separating the *Q* value calculation and selection, with the evaluation and target networks as the double DQN. The evaluation network selects the largest *Q* value. In Section 2, the target network was expressed using Equation (5), which calculates the target *Q* value through the action selected in the evaluation network. Thus, the *Q* value updated using the D3QN is expressed as follows:(22)$$Q\left(s_{t},\;a_{t};\theta^{-}\right)=r+\gamma (V\left(s_{t};\;\theta \right)+A\left(s_{t},\;arg\underset{a_{t+1}}{\max}Q\left(s_{t+1},\;a_{t+1};\theta^{-}\right);\theta \right)\;-\frac{1}{\left|A\right|}\underset{a_{t+1}}{\sum }A\left(s_{t},\;arg\underset{a_{t+1}}{\max}Q\left(s_{t+1},\;a_{t+1};\theta^{-}\right);\theta \right)$$

The *Q* value of the next step was defined by the dueling DQN method as the sum of *V* and *A*. Here, the double DQN method selects the action $a_{t}$ for predicting *A* in the target network, and the corresponding V and A were evaluated in the evaluation network. The new value was updated in the target network.

#### 4.4.2. Proposed Scheduler

The trained model is used as a scheduler for task processing in the SD. The DRL-based scheduler decides the method of processing the input task, as shown in Figure 6. When a task is firstly arrived, the SD’s queue status, battery status, and communication status are collected and transmitted to the DRL-OS to determine the mode for task processing. The modes that can be selected at this point are local, offloading, and drop. Local processing refers to direct processing on an SD. Since all tasks must be processed within a given deadline, if a task is processed beyond the deadline, the task is treated as dropped. However, the difference between the drop and the drop mode beforehand is that latency time and energy consumption for task processing occur in drop by the deadline exceeded. The proposed DRL-OS processes a task using one of three modes: local mode, remote mode, and drop mode, based on SD and MEC status information. Figure 6 is a visualization of the process of DRL-OS, and the process is as follows.

First, when a new task occurs in the SD, the SD collects its own state information and the state information of the MEC, and then selects a processing mode using the pre-trained DRL-OS. The local mode is a mode in which tasks are directly processed in the SD, and both latency and power consumption are large. The remote mode is a mode in which tasks are processed by the MEC. Both latency and power consumption are less than those in the local mode, but the latency may increase because the MEC can receive processing requests from multiple SDs. Finally, the drop mode is a mode in which a task is dropped in advance without attempting to process it when it is expected that the task will not be processed within the deadline. In principle, all tasks must be processed within the specified deadlines, and tasks that are not processed within the deadline are treated as dropped even if the task processing is completed. Therefore, when a task is preemptively dropped through the drop mode, there is an advantage in that no waste of computing resources occurs due to an attempt to process the task.

When the processing mode is determined through the DRL-OS, the SD transfers the task to the queue of the selected processing mode, and in each processing queue, tasks are sequentially processed in the order in which the tasks arrived.

## 5. Experimental Results

In this section, the simulation results are discussed to prove the effectiveness of the proposed DRL-based scheduler. First, the simulation settings are presented. Next, the simulation results with various parameters are analyzed.

### 5.1. Experimental Setting

The basic simulation settings for performance evaluation are as follows. We performed simulation on a computer with a hexacore Intel i7 CPU and a 3.7-GHz, 16-GB RAM processor. The program was written with TensorFlow 2.8.0 in Python 3.9.7. The task size and deadline were uniformly distributed with K∈[50, 450] KB and D∈[100, 300] ms, respectively. The required CPU cycles/bit was 1000. The initial queues of the SD and MEC server were $\delta^{proc}=0$,$\;\delta^{tran}=0$, and $\delta^{MEC}=0$, respectively. A battery with a capacity of 4500 mAh powered the SD. For the simulation, we assumed half the battery capacity (i.e., 2250 mAh). Table 2 lists the key simulation parameters and Table 3 summarizes the hyperparameters with an average score acquired through optimization tuning.

To indicate the simulation result in terms of battery level, the energy consumption was converted into battery consumption and then subtracted from the residual amount of battery charge. We assumed that every device had the same processing performance and were placed at the same distance from the MEC. Thus, every device consumed the same power. Moreover, the mobile device and MEC server used 5G/Wi-Fi communication protocols. When data were transmitted by offloading, one of these types was randomly selected. Identical weights were chosen for the latency and energy of each application task. The simulation scenario contains five wearable devices, 10 SDs, and 1 MEC server.

The occurrence interval of each datum was 100 ms, which was set as one round. Five hundred rounds were set as one scenario, and each scenario was carried out with different simulation parameter settings. An average of 500 rounds was output for each metric.

Three performance metrics were considered to evaluate the method efficiency: drop rate, average residual battery charge, and average latency. The performance of the proposed scheduler was compared with five offloading schemes described below:A local scheme which computes tasks locally by allowing offloading decision parameter without task offloading;A remote scheme which processes tasks by offloading them to an MEC server through a transmission queue instead of processing them locally;A random scheme [41] which performs computation by randomly selecting local and offloading regardless of the task size and network condition;An optimal scheme selects and performs optimal decisions by determining the minimum energy and latency costs;A rule-based scheme [14] is an offloading decision method that considers the queue status to minimize the latency.

### 5.2. Simulation Results and Discussion

#### 5.2.1. Convergence Analysis

Figure 7 illustrates the convergence curve of the average reward for 800 epochs. The *x*-axis represents the number of episodes of the learning process, and the y-axis represents the average reward of all epochs. For 1200 episodes, ε decreased from 0.99 to 0.05, and the discount factor was set to 0.99. According to the graph, the random exploration was terminated at the 1000th episode, and convergence was initiated subsequently.

#### 5.2.2. Impact of Task Size Analysis

This section presents the simulation of the change in the performance of the DRL-OS with task. In all figures, the task size axis outputs the average of 500 rounds performed whenever one input task size was set.

Figure 8 depicts the average of the actions according to the increase in task size. First, the DRL-OS maintains an action of 1.0 and gradually increases to 1.2 from 380 KB. Until 380 KB, tasks were processed by offloading them to the MEC server, but the ratio of the tasks that selected a drop action increased with the task size. This indicates that most tasks were processed by offloading, but the action for the drop was also being executed simultaneously. As shown in Figure 8, Local Only maintains the average action at zero because every task is processed locally. Remote Only maintains the average action value at 1.0, because the tasks are processed by offloading. The random method progresses toward a uniform distribution based on the average action value of 1.0. The optimal scheme maintains the action at 1.0, similar to DRL-OS, and a gradual increase begins at 380 KB. Up to 380 KB, tasks are offloaded to the MEC; however, the decision rate of drop actions are also increased as the task size increases. For most task sizes below 380 KB, the optimal scheme selects the offloading action and if the size exceeds 380 KB, it chooses the drop action to avoid wasting time and battery. The rule-based scheme maintains an average action of 1.0, gradually decreasing below 1.0 from 380 KB. This is because the local action selection increases due to increased task size as the remote processing time become more significant than the local processing time.

Figure 9 illustrates the changes in the average residual battery charge with respect to the task size. First, as expected, the DRL-OS maintained a high average residual battery charge without significant changes, even as the task size increased. The average residual battery charge slightly increased compared with that for Remote Only because certain tasks were dropped after 380 KB. Similarly, Remote Only also maintained the average residual battery charge, which is similar to that of the DRL-OS technique, albeit marginally less than that at 380 KB. This is because Remote Only continues to offload even if the task size increases, which makes it impossible to process tasks. Regarding the Random method, we observed that the average residual battery charge steadily declined. As for Local Only, the average residual battery charge plummeted since the start and stagnated from 240 KB.

Because it is impossible to process within 100 ms from this size locally, the amount of battery charge consumed for 100 ms is always measured. Even if the task size increases, the optimal scheme shows no significant changes in the average residual battery charge because some tasks are dropped at 380 KB for an optimal offloading decision. In the rule-based scheme, the average residual battery does not change until the task size reaches 380 KB and decreases from 380 KB. If the task size exceeds 380 KB, it decreases because some tasks are processed locally.

Figure 10 illustrates the result of drop rate according to the task size. First, the drop rate of DRL-OS is constant from 0 to 300 KB; then, it increases rapidly at 380 KB and then increases linearly. This is because from 380 KB, a few tasks are dropped as tasks begin to accumulate in the queue. However, Figure 10 shows that the ratio of dropped tasks is significantly less than those of the other schemes.

Remote Only also exhibited a drop rate similar to that of DRL-OS when the task size was small; it exponentially increased from 380 KB and reached 1.0. In other words, all tasks were processed as being dropped. This is because Remote Only unconditionally offloads regardless of the state of the MEC server, and the MEC server unconditionally attempts to process it regardless of the possibility. Therefore, while resource consumption due to the processing attempt remained the same, almost all tasks breached the deadline. Consequently, the corresponding tasks were processed as being dropped. In addition, as a task occurs every 100 ms, tasks awaiting processing are accumulated in the MEC server’s queue as the number of tasks exceeding the deadline increases.

Therefore, the method is caught in a tedious cycle of attempting to process the task, then attempting to process the dropping of tasks, and finally processing the failed iterations. Thus, no amount of processing can be completed through offloading. In the Local Only method, the drop rate remains 0 for a small task size, but rapidly increases from 320 KB. This leads to a processing failure due to a cycle similar to that of the Remote Only, as Local Only cannot process tasks within the deadline from 230 KB. In the random scheme, the drop rate is maintained below 0.4 from the initial stage to 230 KB. From 230 KB, however, it increases beyond 0.6; we confirmed that more than half of the tasks were dropped. Figure 10 demonstrates that the drop action is in progress through the average action value of the random scheme. The optimal scheme maintains the drop rate at 0 until the task size exceeds 380 KB. From the task size of 380 KB, the drop rate increases stepwise, and the increase is smaller than the DRL. The reason is that in the case of optimal scheme, the task that cannot be processed is accurately identified and dropped, so there is no excess of the deadline due to queue accumulation at all. In the case of DRL, there is a difference in the drop rate from the Optimal because the selection was inconsistent with the Optimal. For small task sizes, the rule-based scheme shows drop rates similar to that of DRL-OS and the optimal scheme. Then, it exponentially increased to 0.9 from the task size of 380 KB. This leads to processing failures similar to that observed in the Remote Only as the Rule-based scheme cannot process tasks within the deadline.

Figure 11 shows the average latency results with respect to the task size. The average latency in Local Only increases rapidly from start to task size of 230 KB and is not calculated when it exceeds 230 KB. This is because the latency calculation is performed only for the task that has been successfully processed. If the task size exceeds 230 KB, the Local Only cannot process the task within the deadline. Moreover, Local Only consumes time and battery regardless of whether tasks are dropped because it attempts to process tasks even after the deadline. Therefore, when the deadline is breached, the task is processed and then dropped. The random scheme performs parallel processing using local and remote when the task size is small. Therefore, it has a small latency compared to Local Only. If the task size exceeds 230 KB, it has only latency due to remote processing because it cannot be processed locally. However, the queuing delay is short because the task is processed once every 300 ms, that is, three times the generation period on average. DRL-OS, Remote Only, Optimal, and rule-based have almost the same average latency because tasks are processed remotely up to a task size of 380 KB. In addition, the queuing delay is slightly higher than that of random because the average period of accumulating tasks in the queue is approximately 100 ms, which is shorter than the average period of random. If the task size exceeds 380 KB, the rule-based parallelizes local and remote, unlike the other three schemes. However, it takes only processing time by offloading because local processing is impossible due to the excess of the deadline. As more remote processing is attempted compared to Optimal or DRL-OS, a relatively long latency due to queuing occurs.

When the task size reaches 450 KB, most of the tasks are dropped due to exceeding the deadline. Moreover, the latency is lower than that at 445 KB because the latency is calculated only for tasks processed successfully at the beginning of the round. However, as the drop rate is almost 100%, it is not a statistically significant value. For Remote Only, all processing is attempted remotely, but unlike rule-based, deadline exceeding due to queuing delay occurs from the initial round. In addition, the number of successfully processed tasks is small, and as it is the latency for the tasks processed in the beginning, it is shorter than that of the rule-based. As Optimal selects a drop action if it is not a task that can be processed, the other tasks are processed well except for the task determined to be dropped. Notably, the unprocessable tasks are dropped every task generation cycle and the queuing delay is reduced; therefore, the latency is shorter than that in the other three schemes. DRL-OS parallelizes drop and remote processing similar to Optimal. However, as it does not determine unprocessable tasks similar to Optimal, and some tasks are further delivered to the remote, resulting in latency due to queuing delay.

#### 5.2.3. Impact of Deadline Analysis

In this section, the effect of a variable deadline on the performance of the scheduler is discussed. A smaller deadline implies a higher sensitivity to delays. Thus, the performance is verified through various indices with variable deadlines, and the results are analyzed. (These results must be added to the task size). In the simulation, the occurrence interval cycle of each data was 100 ms, and the task size was fixed to 80 KB. Deadline D_i increased variably by 10 ms from 100 to 300 ms.

Figure 12 illustrates the results of the average action with respect to the deadline. The average action of DRL-OS was greater than 1 until 150 ms. This indicates that the offloading and drop modes were selected simultaneously. However, in cases where the processing time exceeds the deadline (150 ms), the average action decreases to 0.98, which indicates that the tasks are remotely processed up to 150 ms, while a few are processed as dropped if they demand a considerably longer time than the deadline. However, the average action decreases below 1.0 if the deadline exceeds 150 ms, and then the tasks are processed locally and remotely in parallel. Near the deadline of 150 ms, a considerable volume of tasks are selected for local processing owing to their relatively short deadlines. As the deadline increases, the processing of offloading increases very slightly again. However, the change is minuscule, and the performance is virtually the same. The average action of the optimal scheme was greater than 1 until the deadline reached 170 ms. This shows that the drop and offloading actions are selected. If the deadline exceeds 160 ms, the average action decreases to 0.98, as some tasks are processed locally. Moreover, when the deadline is 100 ms, Optimal has a more significant average action than DRL-OS. This means that more drop actions are selected. In the Rule-based scheme, the tasks are processed locally and remotely in parallel, and the selected actions are almost the same in all rounds because the task size does not change. The performance of Remote Only did not change because the average action is 1.0, and all tasks were remotely processed regardless of the deadline. The average action of the random method has an action value of approximately 1.0 because of the uniform selection of the three modes. Local Only has an average action value of 0 because it only selects local processing for the task.

Figure 13 depicts the average residual battery capacity with respect to the deadline. At 150 ms, the average residual battery capacity of the DRL-OS decreases, and does not undergo any significant changes even when the deadline is changed. Offloading and drop actions were selected in parallel from 100 to 150 ms, while offloading and local actions were paralleled from 150 ms, at which no drop actions were selected, thus slightly increasing the battery consumption.

However, because most tasks were processed in the remote mode, the average battery level remained at 2200 mAh despite a few tasks being processed in the local mode. As the deadline was extended, the proportion of the tasks processed in the remote mode increased, such that the residual battery capacity also increased. In the optimal scheme, at the deadline of 170 ms, the average battery charge drops to 2200 mAh and remains constant. This is because local processing is increased at the deadline of 170 ms. However, the longer the deadline, the more tasks the remote can handle, and therefore, the average battery charge increases slightly. As the rule-based scheme only considers time, it has the same average residual battery charge regardless of the change in the deadline. In the random scheme, unlike the DRL-OS and remote scheme, the battery consumption increased due to the proportion of tasks processed in the local mode because the capacity was uniformly selected from the local, remote, and drop action. In this aspect, Local Only records the lowest battery level because it forces only the local mode that consumes the most battery power.

Figure 14 illustrates the result of the drop rate of the task with respect to the deadline. In DRL-OS, an average of approximately 40% drop occurs before the deadline of 160 ms, at which the action value is ≥1.0, as shown in Figure 12. However, if it exceeds 160 KB, most of the tasks are processed successfully. The random mode exhibits a drop rate of approximately 66% at the deadline of 100 ms. The rate remains unchanged until 170 ms, even with a deadline extension. The deadline is shortened from 180 ms, and subsequently a drop rate of 40–50% occurs. In the Optimal scheme, a small drop occurs initially in the deadline. However, the tasks are processed without drops from the deadline of 180 ms because the number of tasks that needed to be locally processed increased. The rule-based scheme has a task drop rate of 0.7 from the deadline of 100 ms.

However, from the deadline of 180 ms, the tasks can be processed without drops. This is because, when the deadline is short, local processing is not possible, and remote processing is only partially possible. However, if the deadline is more than 180 ms, all tasks can be processed in time. Remote Only exhibits a high drop rate in all deadlines because the SD generates data of 400 KB in every round. As mentioned earlier, Figure 10 confirms the 400 KB task size and the high drop rate at a deadline of 100 ms. However, as the deadline is further extended, the number of tasks that can be processed increases. This confirms that the drop rate gradually decreases. Local Only exhibits a drop rate of nearly 100% because it cannot process most tasks within the long and short deadlines. Although a minimum number of tasks were processed after 180 ms, almost all were dropped as the number was extremely small.

Figure 15 illustrates the average latency result with respect to the deadline. Local Only shows that every input task is dropped with a deadline of 100 ms to 180 ms. If the deadline exceeds 180 ms, the task can be processed within the deadline, and the processing time can be calculated. As the deadline increases, the number of tasks that can be processed increases; therefore, the latency increases as well. The average latency of Remote Only gradually increases because the number of tasks to be offloaded increases proportionally with the deadline. The random scheme generally has a low latency. This is because only 33% of the total generated tasks from 100 ms to 170 ms are periodically processed remotely. From the deadline of 180 ms, the average latency increases because some tasks assigned for local processing are also processed. In the optimal scheme, the latency gradually increases because the number of tasks to be processed increases proportionally to the deadline, similar to the Remote Only scheme. However, it has a steeper increment to the Remote Only because more tasks are processed. In the rule-based scheme, when the deadline is small, the drop rate and latency increase due to the accumulation of remote queues. However, if the deadline is large, task processing is requested to the side with the shortest latency without any other consideration so that it has a consistent latency.

DRL-OS only handles tasks remotely when the deadline is between 100 ms and 140 ms. Then, after 150 ms, some tasks can be processed locally, and there is an effect of reducing the latency by parallel processing. When the deadline is 170 ms or more, the latency gradually increases as the ratio of the remote increases. Moreover, the local processing increased as most data were processed remotely, and the model risks breaching the deadline if the number of tasks to be offloaded is increased further. Therefore, the task was processed locally so that it would not be dropped. The cost of local processing would have been smaller than the cost of dropping. The offloading mode, which incurred a smaller cost, was selected as the energy cost was higher between the delay time cost according to remote processing and the local energy cost. Therefore, the farther the deadline, the more significant the proportion of the remote mode because the waiting time in the remote mode becomes longer and the processing time increases. Nonetheless, it had a smaller value compared with the Remote Only or Local Only modes.

Although this study presents a comprehensive analysis of the computation offloading scheduling problem from the perspective of SDs, further work is necessary to compare the results with previous studies (for instance, comparison with offloading scheduling from the perspective of MEC). In addition, statistical hypothesis tests conducted on the proposed scheme and the measurements of the remaining schemes confirmed that there were statistically significant differences in the measurement results obtained through the applied metric. The results of this section confirmed that the DRL-OS method outperforms existing models in various aspects with variable deadlines

## 6. Conclusions

This study proposed the DRL-OS, which uses energy balance to select between methods based on local computing, offloading, or dropping for performing a task. We investigated the computation task offloading problem for computation-intensive and delay-sensitive tasks in an MEC environment. We designed the DRL-OS using the D3QN, which enabled offloading decision making to minimize costs in terms of both delay and energy of the SD. We considered deadlines in addition to latency and energy consumption and formulated the optimization scheduling strategy that considered the benefits of reducing the task dropping of applications. To analyze the performance of the scheduler, three conventional offloading methods, i.e., local, remote, and random, were simulated in a scenario where the task size and deadline varied. The simulation result confirmed that the proposed algorithm guaranteed a higher battery level, a lower average latency, and a lower task drop ratio than other methods.

However, this study did not consider the separate processing of tasks. Hence, a scheduling method considering the same needs to be researched. The task segmentation feature can be significant for the high load level of an SD with limited computing resources and battery. Therefore, hierarchical multi-tier partial offloading reflecting this feature is a viable future direction of this research.

## Figures and Tables

**Figure 1 sensors-22-09212-f001:**
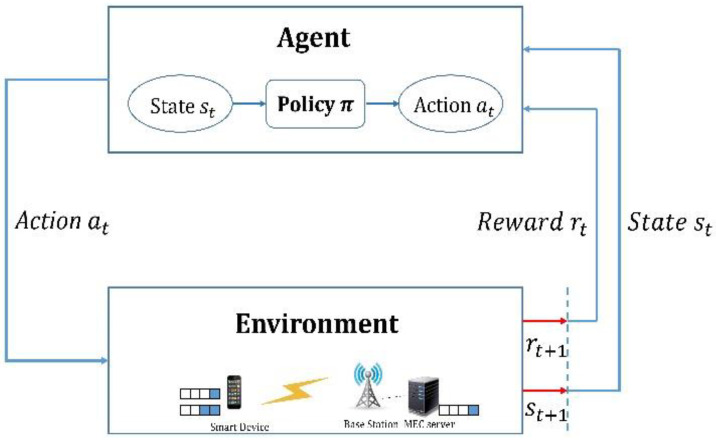
Reinforcement Learning.

**Figure 2 sensors-22-09212-f002:**
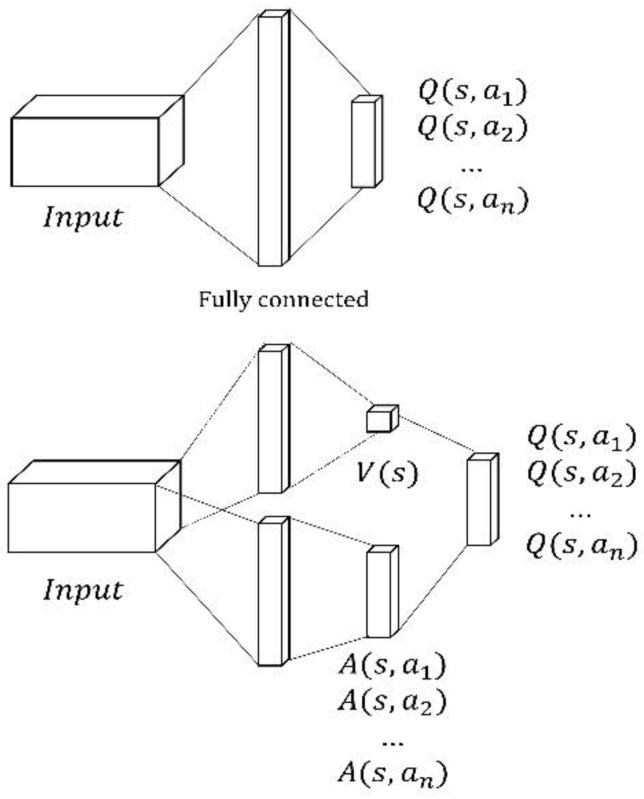
Deep Q-Network (**top**) and Dueling Deep Q-Network (**bottom**).

**Figure 3 sensors-22-09212-f003:**
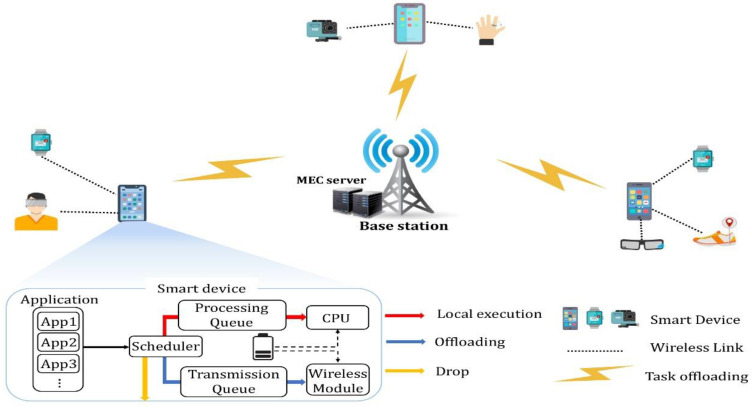
Scenario of the MEC system Model.

**Figure 4 sensors-22-09212-f004:**
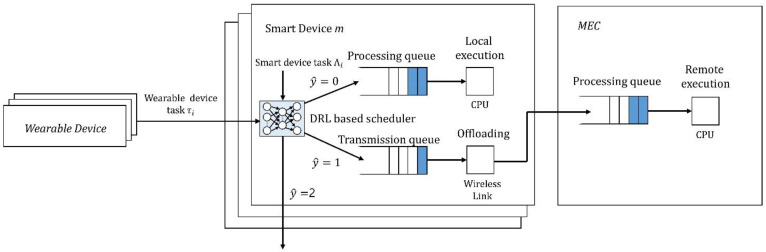
Illustration of the structure of the smart device with a DRL-based scheduler.

**Figure 5 sensors-22-09212-f005:**
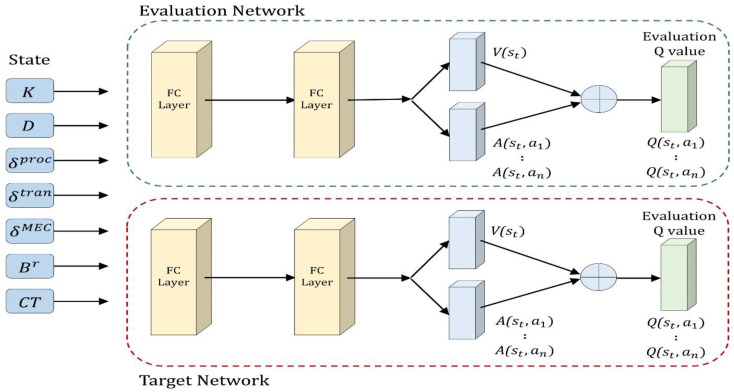
Network architecture of D3QN.

**Figure 6 sensors-22-09212-f006:**
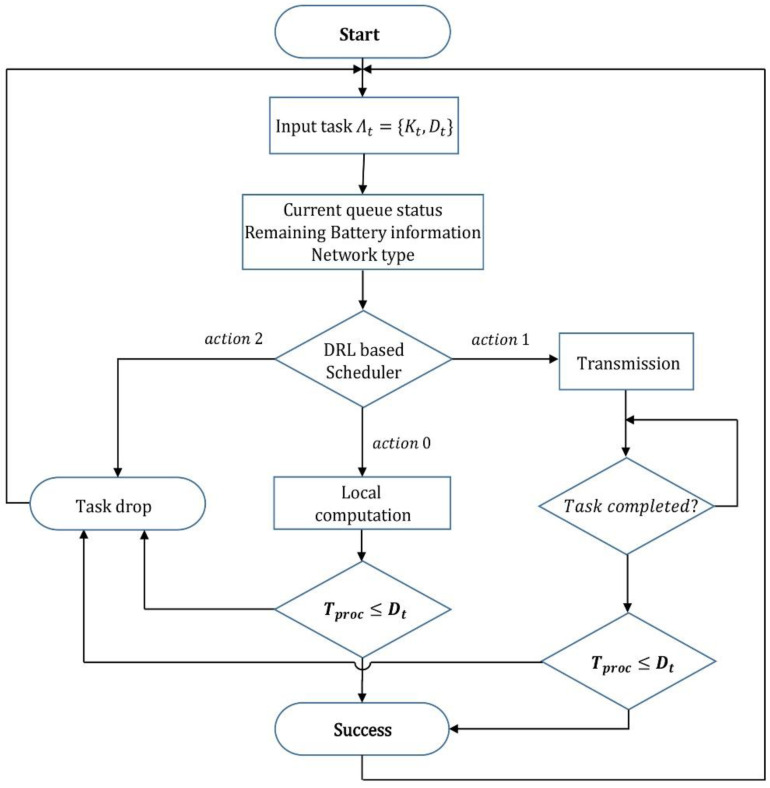
Scenario of the MEC system.

**Figure 7 sensors-22-09212-f007:**
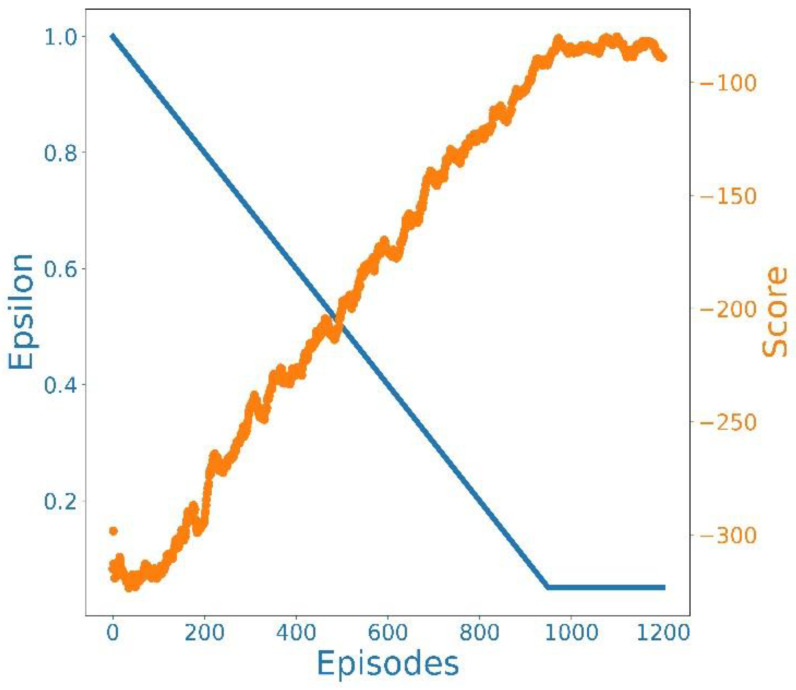
Convergence of the D3QN-based Scheme.

**Figure 8 sensors-22-09212-f008:**
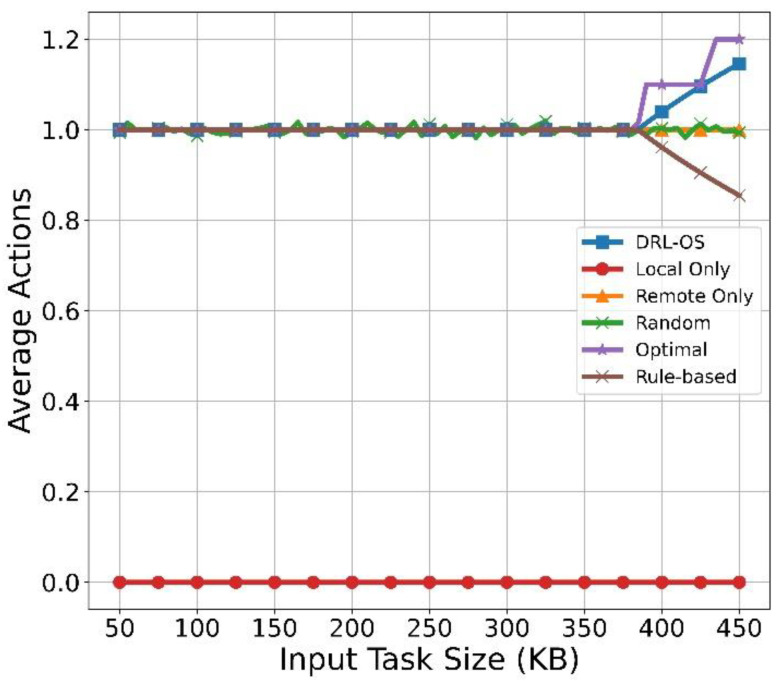
Task average action vs. input size.

**Figure 9 sensors-22-09212-f009:**
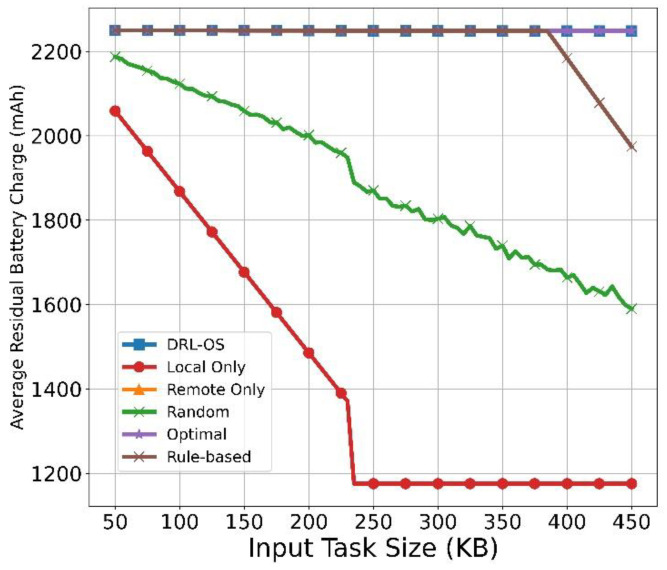
Average residual battery vs. input task size.

**Figure 10 sensors-22-09212-f010:**
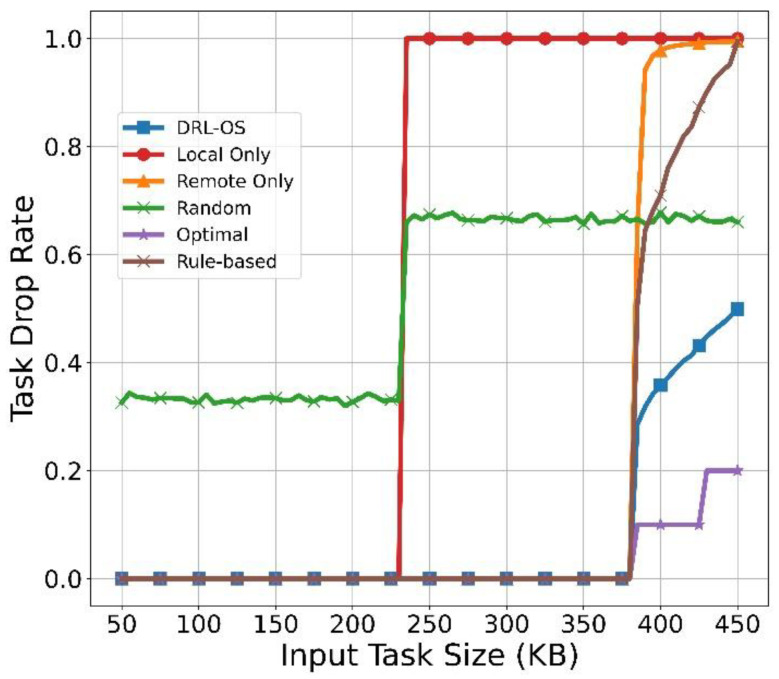
Task drop rate vs. input task size.

**Figure 11 sensors-22-09212-f011:**
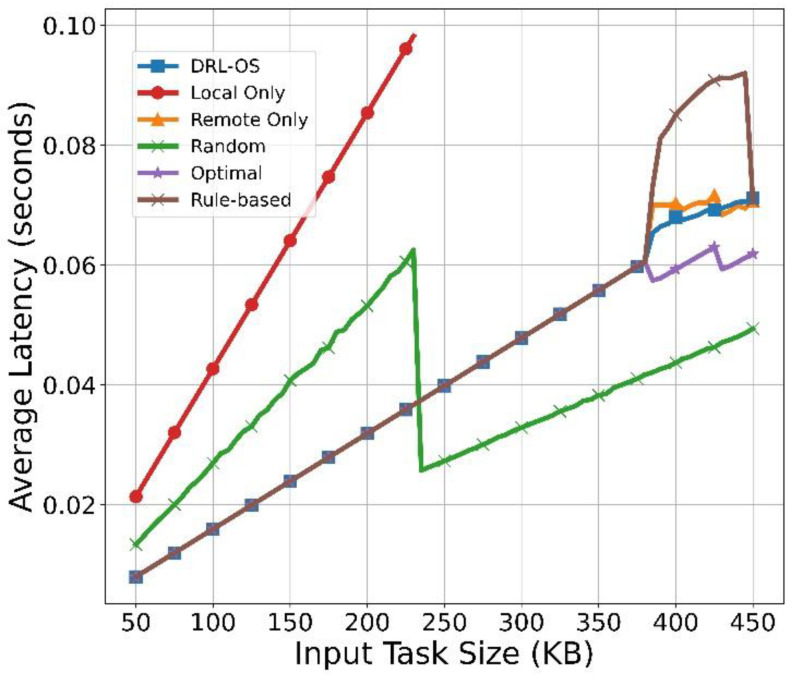
Average latency vs. input task size.

**Figure 12 sensors-22-09212-f012:**
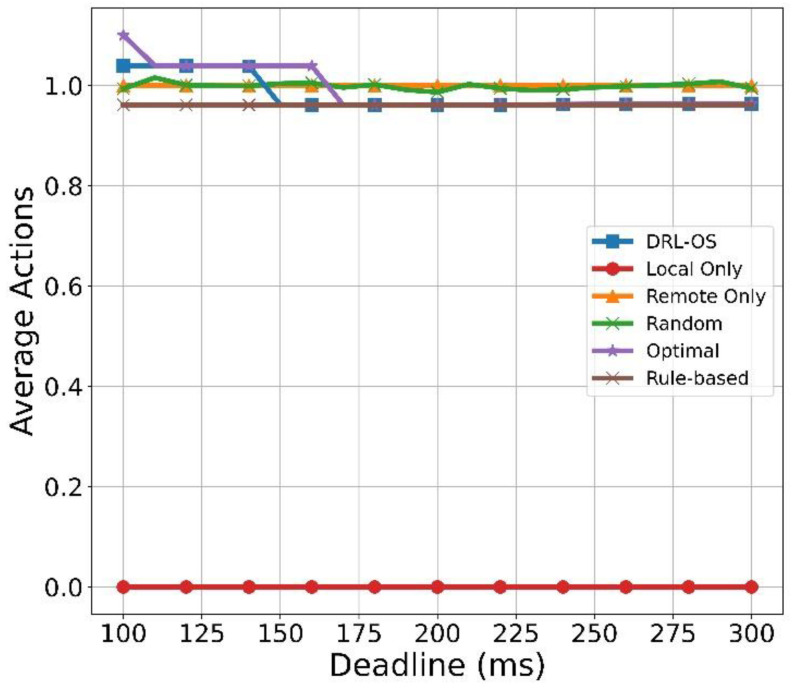
Average action vs. deadline.

**Figure 13 sensors-22-09212-f013:**
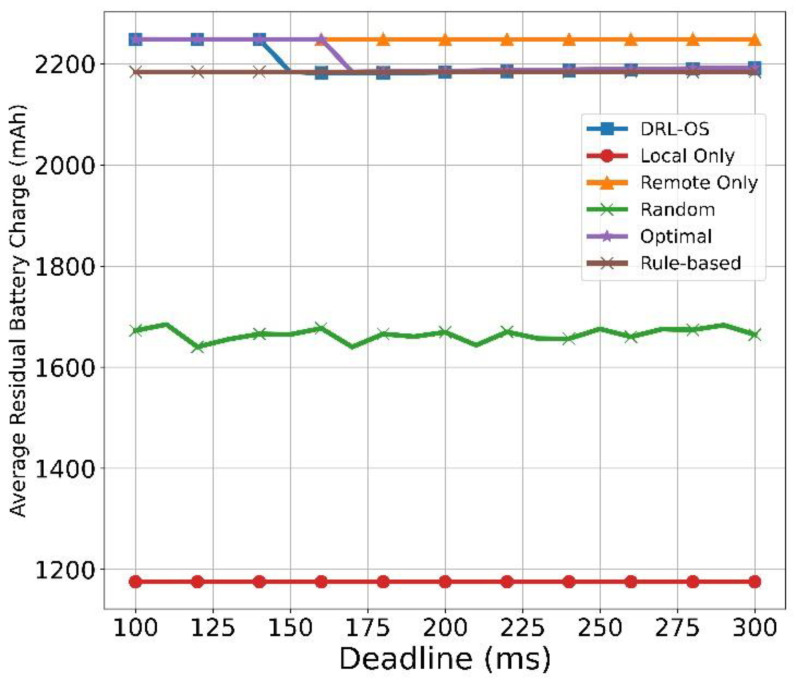
Average residual battery vs. deadline.

**Figure 14 sensors-22-09212-f014:**
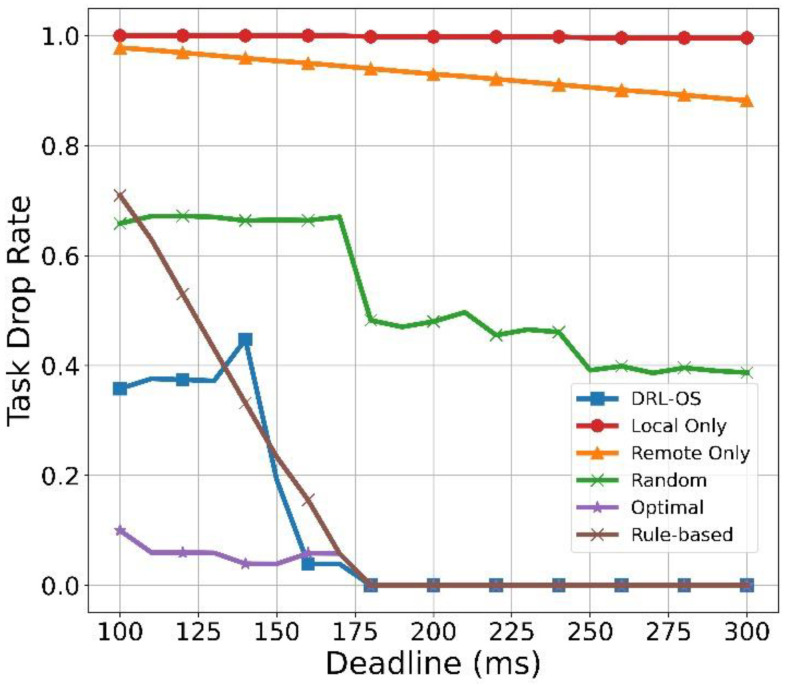
Task drop rate vs. deadline.

**Figure 15 sensors-22-09212-f015:**
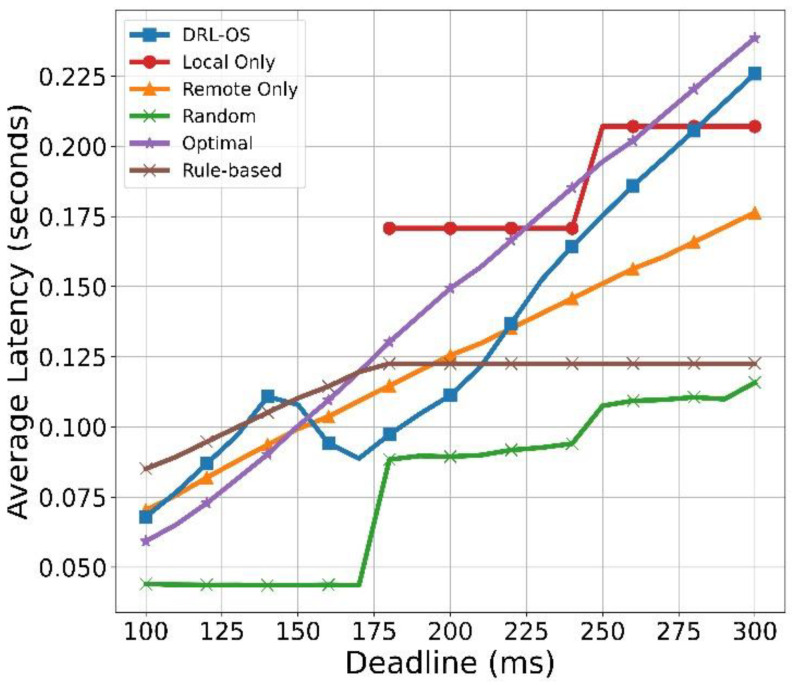
Average latency vs. deadline.

**Table 1 sensors-22-09212-t001:** Notations.

Notation	Description
*M*	A set of SDs
$core^{\mathrm{loc}}$	Number of CPU cores in the device
*K*	Size of the task
*D*	Deadline of the task
cycle	Number of CPU cycles required to process the task
$\delta^{proc}$	Processing queue of devices
$\delta^{tran}$	Transmission queue of devices
$L_{i}^{loc}$	Processing time of the task $\Lambda_{i}$ by local computing model
${ℱ}^{loc}$	Computational capability of SD
$P^{loc}$	Computing power of device
$E^{loc}$	Energy consumption of the task $\Lambda_{i}$ by a local computing model
$L_{i}^{tr}$	Time for transmission $\Lambda_{i}$ from the SD to the MEC server
$DataRate^{CT}$	Transmission rate of the SD
$L_{i}^{MEC}$	Processing time of task $\Lambda_{i}$ by the MEC server
$\delta^{MEC}$	Processing queue of MEC
${ℱ}^{MEC}$	Computational capability of MEC
$L_{i}^{ofl}$	Total processing time when task $\Lambda_{i}$ is processed by offloading it to the MEC server in the SD
$E_{i}^{ofl}$	Energy consumption of task $\Lambda_{i}$ by computational offloading
$P^{tr}$	Transmission power of the SD
$C_{t}^{loc}$	Cost of local computing
$C_{t}^{ofl}$	Cost of processing by offloading
ω	Weighting coefficients

**Table 2 sensors-22-09212-t002:** Simulation Parameters.

Parameters	Settings
5G data rate	100 Mbps
Wi-Fi data rate	200 Mbps
Transmission Power, $P^{tr}$	24 dBM [36]
Input task size, K	50–450 KB Default: 400 KB
Required CPU cycles per bit	1000 CPU cycles/bit
Local computational capability, ${ℱ}^{loc}$	$\left[2.34\times 10^{9}\right]$ CPU cycles/s
MEC computational capability,$\;{ℱ}^{MEC}$	$\left[2.4\times 10^{9}\right]$ CPU cycles/s
Local CPU core	8
Edge node CPU core	128
Weight factor	0.5

**Table 3 sensors-22-09212-t003:** Hyper Parameters.

Parameters	Value
Learning rate	0.0005
Discount factor	0.99
Min epsilon	0.05
Init epsilon	0.99
Epsilon decay	0.00002
Batch size	64
Episodes	1200

## Data Availability

Not applicable.

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
