# Peer review of "DRL-OS: A Deep Reinforcement Learning-Based Offloading Scheduler in Mobile Edge Computing"

_sensors, 2022, doi:10.3390/s22239212_

Round 1
Reviewer 1 Report
The manuscript has merits and can be accepted for publication upon considering the following comments.
1. Literature review should be incorporated in a separate table so that repetition can be minimized.
2. The whole manuscript can be proofread by an English Expert before the next revision.
3. The abstract is a bit vague, please make it more clear by indicating the actual problem and devising the solution.
Reviewer 2 Report
The paper reports the development of a deep reinforcement learning-based offloading scheduler to select a method for performing a task from local computing, offloading, or dropping by considering the energy balance. The scheduler performance is measured in terms of the average battery level, drop rate and average latency. The performance of the proposed approach is compared against five commonly used offloading schemes. The experimental established results confirmed that the proposed algorithm outperforms the considered offloading methods, guarantying a higher battery level, a lower average latency and a lower task drop ratio.
This paper deals with an interesting subject and the proposed method seems to be sound. The experimental results show that the method developed based on D3QN works better than the rest of the tested methods. The paper is well written, but some further explanations and some minor changes could be added to make it clearer.
- Explain the updating rule (3).
- The appearance of fig. 3 could be improved
- Explain the parameters in (15)
- The appearance of fig. 6 could be improved .
Reviewer 3 Report
The paper provides a study above using a reinforcement learning mechanism to implement an offloading scheduler in mobile edge computing. The paper is interesting. However, some issues need to be solved:
1. All the assumptions need to be clearly stated. Please provide the list of all the considered assumptions (e.g. a bulleted list) and their impact (e.g. on generality of the problem, on real-life scenarios, etc.).
2. Why the authors include the term “Mobile Edge Computing” in the title of the paper? By combining the assumption written in lines 269-270 (“SDs are connected to one MEC server”) with the one from lines 270-271 (“the SDs were assumed to be fixed or have very low mobility”) we may conclude that the term “Mobile” is useless. Please explain.
3. How the sentence from lines 92-93 (“Each task can be…”) match with the “more realistic scenario” (see line 88)? It seems that the mentioned sentence is an assumption that excludes multitask scenarios.
4. The communication delay needed to offload the tasks is considered to be zero?
5. The time to schedule the tasks to be offloaded is considered to be zero?
6. The energy used by scheduler is considered or not?
7. Please specify where exactly the offloading scheduler is intending to be run.
8. The task dropping procedure, at SD level or at edge level, is not explained in details. On what basis this process is done? For example, in line 99 it is written “If it is difficult to process the task, the SD drops the task without processing it” which is confusing (mainly the term “difficult”). How can we estimate if a task is difficult or not before trying to process it?
9. Line 275: why do we need an intermediate node (i.e. base station (BS)/access point (AP)) to transfer the tasks from SDs to MEC?
10. Is it important to consider two communication types (see line 344) if the time needed for communication is not explicitly considered?
11. Line 433 and 437: the sequence “task is processed in the drop mode” is strange. A task is either “processed” or “dropped”.
12. Line 435: the conclusion ”Therefore, the probability of requiring this process is low.” Is not well explained.
13. Table 2: the “local” and “MEC” computational capabilities are very close (2.34x10^9 vs 2.4x10^9). In this case the difference between “local” and “edge” lies only in the number of cores?
14. How MEC handles multiple requests from SDs?
Round 2
Reviewer 3 Report
The paper has been substantially improved. All my comments and concerns were appropriately solved.